# Inducing Performance of Commercial Surgical Robots in Space

**DOI:** 10.3390/s23031510

**Published:** 2023-01-29

**Authors:** Timothy Sands

**Affiliations:** Department of Mechanical and Aerospace Engineering, Cornell University, Ithaca, NY 14853, USA; tas297@cornell.edu

**Keywords:** space robotics, surgical robots, autonomous medical robots

## Abstract

Pre-existing surgical robotic systems are sold with electronics (sensors and controllers) that can prove difficult to retroactively improve when newly developed methods are proposed. Improvements must be somehow “imposed” upon the original robotic systems. What options are available for imposing performance from pre-existing, common systems and how do the options compare? Optimization often assumes idealized systems leading to open-loop results (lacking feedback from sensors), and this manuscript investigates utility of prefiltering, such other modern methods applied to non-idealized systems, including fusion of noisy sensors and so-called “fictional forces” associated with measurement of displacements in rotating reference frames. A dozen modern approaches are compared as the main contribution of this work. Four methods are idealized cases establishing a valid theoretical comparative benchmark. Subsequently, eight modern methods are compared against the theoretical benchmark and against the pre-existing robotic systems. The two best performing methods included one modern application of a classical approach (velocity control) and one modern approach derived using Pontryagin’s methods of systems theory, including Hamiltonian minimization, adjoint equations, and terminal transversality of the endpoint Lagrangian. The key novelty presented is the best performing method called prefiltered open-loop optimal + transport decoupling, achieving 1–3 percent attitude tracking performance of the robotic instrument with a two percent reduced computational burden and without increased costs (effort).

## 1. Introduction

**Figure 1 sensors-23-01510-f001:**
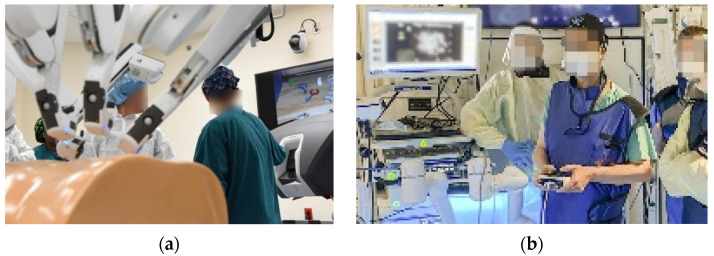
(**a**) Air Force surgeon receives a briefing from a nurse on robotic surgery capabilities inside the robotics surgery clinic at Keesler Medical Center, Miss., 16 June 2017 [1]. (**b**) VA Medical Center uses new Thoracic Oncology robotic arm to diagnose lung cancer in veterans [2]. U.S. Department of Defense photographs and imagery, unless otherwise noted, are in the public domain [3]. Images courtesy U.S. Department of Defense [4].

A very recent review of surgery in space (e.g., the International Space Station, planned Lunar Gateway, etc.) indicates the high utility of autonomous capability to perform certain procedures (e.g., Figure 1) without human-intervention or surgeon-assisted robotic surgery [5] akin Figure 2. Surgical diseases, such as appendicitis and cholecystitis, can occur without warning signs [6], and virtually any unpredictable event that may occur outside the terrestrial orbit is a risk for the individual crew member, the whole crew, and the mission. The need for highly accurate instrument pointing whilst free-floating in space as depicted in Figure 2 is paramount to ensure patient safety. 

Historically, screening and testing nodules in less accessible areas of the lungs has been difficult. Using a procedure called a bronchoscopy, physicians manually ran a tube with a small camera inside, down a patient’s throat and into the lungs. From there, they maneuvered the camera or instruments to see and biopsy suspicious nodules [2].

### 1.1. Context of the Problem: Medical Procedures Free-Floating in Space

Ball, et al. explores the potential benefits and defined risks associated with prophylactic surgical procedures for astronauts including, but not limited to, appendectomy [9] and cholecystectomy [10]. Haidegger highlighted the need in [11] for space robotic surgical systems to perform basic surgical procedures, such as suturing; they should provide diagnostic instrumentation and interpretation for ultrasound, computed tomography-scan (CT-scan) or magnetic resonance imaging (MRI). The following year, Heidegger [12] reviewed the existing autonomous capabilities of surgical robots and investigated the major barriers of development presented by the lack of autonomy *benchmarks and standards* (emphasis added). As early as 2008, Hamilton, et al. [13] highlighted the need for autonomy in reaction to severe traumatic injury during long duration spaceflight [14].

### 1.2. Manipulation of Very Light, Highly Flexible Medical Robotic Instruments

#### 1.2.1. Challenges in the Manipulation for Surgical Space Applications

Dynamic coupling between the arm and base is particularly important. Inclusion of fictional forces (e.g., Coriolis force, etc.) is another challenging aspect since these forces couple rotational states nonlinearly. Multi-body treatment of controls-structural interaction is another key challenge. While the third challenge is not addressed in this work, the first two of these challenges are addressed in the context of fuel–minimization. Abruptly changing mass is a mandatory topic, so approaches presented here necessarily incorporate adaptive or learning capabilities in response. In order to address the challenge, instrument sensors and feedback are necessary. 

Evaluation of deleterious effects of randomly varied instrument properties fusing low-quality, noisy sensors utilized by recent methods developed from systems theory.Evaluation applied to realistic systems without neglecting nonlinear coupling effects of rotational transport theorem.


#### 1.2.2. How the Challenges Were Addressed in the Past and Literature Gaps

Flexible control meets optimization: Recalling the need for highly accurate instrument pointing whilst free-floating in space to ensure patient safety, state of the art methods for highly flexible space robotic systems were reviewed in [15] suggesting high efficacy of classical approaches where command shaping was limited to simple, single sinusoids. Meanwhile, the recently published reference [16] suggested substantial performance improvements were possible discarding the classical approaches in favor of robotic system optimization by so-called “systems theory”, an amalgamation of half-a-dozen or so mathematical necessary conditions yielding an analytically solvable boundary value problem. Both Refs. [15,16] illustrate how rigid body treatments are foundational and extended to highly flexible space robotic systems by application of rigid body treatments of a single body to multi-body systems, where medical instruments are mathematical concatenations of rigid body treatments at multiple nodes of a flexible body (embodied in the finite element method). 

#### 1.2.3. Advanced Control Methods for Manipulation

Systems theory amplified with feedback of noisy sensors: Inspired by the suggestion, reference [17] espoused a treatise on disparate formulations of the optimization problem to reveal substantial differences in the robotic system’s performance, depending upon the formulation utilized. Similar to the approach taken in [17], this manuscript compares disparate approaches for efficaciously controlling the rigid body mode of the robotic surgical instrument, while leaving for the sequel appending flexible modes using amplifying techniques suggested in [15,16], particularly the results flowing from systems theory. Systems theory has been successfully applied to a myriad of autonomous operations in space [18,19,20,21,22,23] and will be leveraged in the research presented here, while the resulting optimal control may prove less valuable than the complete solution provided by the rigorous approach. 

The research described in this manuscript borrows attitude pointing techniques from free-floating spacecraft and applies such methods to attitude tracking of rigid space robotic surgery instruments which need to react to sudden changes in system properties (e.g., mass and mass moments) and fusion of realistically noisy instrument sensors. Stemming from earlier work in non-linear adaptive methods [18], which were experimentally validated in [19], Smeresky and Rizzo produced their award-winning article [20] introducing the seminal articulation of analytic, optimal feedback facilitating fully autonomous operations in idealized situations, where their proposal illustrated roughly three-orders-of-magnitude improved accuracy of position (applicable to the necessarily delicate position of a surgical robotic arm) as depicted in Figure 3. Their inspiring results (neglecting random parameter variation and sensor noise) manifest improvement from the order of milliradians with classical and optimal methods to microradians with their proposed approach. The current state-of-the-art moves away from feedback tuning (eliminated by Smeresky and Rizzo) and instead towards optimization of trajectory generators. 

Comparative benchmarks for this research are declared amidst these various methods, and in such accordance, Sandberg [21] just produced a comparative piece investigating autonomous trajectory generation (positions, velocities, and accelerations) applicable to space robotic arms and surgical instruments where the comparative benchmark adopted in this manuscript is so-called classical velocity control. Just a few months ago, Raigoza [22] enhanced Sandberg’s work by adding autonomous collision avoidance capabilities embedded in the autonomous trajectory generators applicable to forbidding medical instruments from maneuvering into positions that endanger patients. Most recently, Wilt critically evaluated Raigoza’s methods amidst utilization of poor-quality, noisy sensors [23], seemingly providing an initial validation of the work. This present manuscript leverages these recent advancements and seeks to extend efficacy, controlling realistic space robotic surgical instruments by combining feedback and pre-filtering, designed using both classical and modern optimal methods. The extended efficacy is presented to address gaps in the literature and propose new innovations.

**Table 1 sensors-23-01510-t001:** Lineage of the reviewed literature ^1^.

Topic	Literature
Robotic surgery training program	[1]
Example medical robotic system	[2]
Robotic surgery in space	[5,6]
Spinal ultrasound in space	[7]
Long duration missions	[9,10,11,14]
Need for autonomy	[12,13]
Legacy: classical flexible control	[15]
Contemporary: optimal flexible control	[16]
Contemporary: nonlinear optimal control	[17]
Contemporary: nonlinear adaptive control	[18,19]
State of the art: learning control	[20]
State of the art: autonomous trajectories	[21,22]
State of the art: deterministic AI	[23]
Novel: systems theory	[24,25,26]

^1^ Table 1, Table 2, Table 3, Table 4 and Table 5 are distributed throughout the manuscript to increase the ease of reading, while a combined, master table of definitions is included in the appendices.

Table 1, Table 2, Table 3, Table 4 and Table 5 are distributed throughout the manuscript to increase the ease of reading, while a combined, master table of definitions is included in Appendix B.

### 1.3. Literature Gaps and Innovations Proposed Aligned with the Literature Review

Control of floating robot systems may be described as comprising moving base, controlled base, kinematic singularities only, fuel consumption, and unlimited workspace [24]. The innovations presented here align with minimization of fuel consumption for controlled base situations. While [24] focused on kinetics derived from Lagrange–Euler equation, the present contribution utilizes Chasles’ theorem and implementation of Pontryagin’s methods [25] embodied in so-called “systems theory” [26]. 

Systems theory utilizes necessary conditions of optimality (with promises of improved performance). In general, contemporary literature utilizes systems theory to develop optimal (open loop) controls, while novel literature seeks to utilize other optimal outputs of systems theory (e.g., instrument motion trajectories). While the predominant literature is for idealized systems, evaluation of methods applied to realistic systems are proposed: Evaluation of deleterious effects of randomly varied instrument properties fusing low-quality, noisy sensors utilized by recent methods developed from systems theory.Evaluation applied to realistic systems without neglecting nonlinear coupling effects of rotational transport theorem.Direct comparison of several disparate options against a declared comparative benchmark(s). Rather than presenting outcomes of performing the diagnostic/surgical procedure, challenges are highlighted along with the advantages and disadvantages of the methods expressed using canonical figures of merit.Comparisons were made using several standard figures of merit, including means and standard deviation of instrument pointing errors, robot movement costs (effort), and computational burdens of each disparate method evaluated.Based on combined accuracy and precision, robustness, and cost (effort), identification of the best selection of approaches is presented to induce improved performance on commercially purchased medical robots lacking the ability to replace the indigenous electronics.

## 2. Materials and Methods

Robotic systems’ mechanics are well studied beginning (and often concluding) with rigid body treatment of double integrator without transport theorem, neglected as high order nonlinear coupling terms. The well-studied fields are invoked in the application of modern systems theory and the approach expressed in this section begins with simplified treatment of surgical robotic system mechanics, followed by articulation of realistic transport theorem, elaborating terms often neglected in initial simplified treatments. Next, innovative classical methods for robot control are introduced but not declared the benchmark approach. Instead, systems theory is used to develop mathematically optimal designs (for idealized systems), and those designs are combined with the classical approaches to declare comparative benchmarks. Eleven disparate competing paradigms are introduced, including basic design approaches based off simple linear avatars of the more complicated, nonlinear coupled system equations. These avatars are useful as theorical benchmarks based on idealized systems. Approaches based on the full, nonlinear differential equations are next presented. Subsequently, in Section 3, the Results, all the approaches are simulated in SIMULINK^®^, and the results are compared amidst randomly varying system mass and mass moments of inertia and fusion of low-quality, noisy state and rate sensors. 

### 2.1. Mechanics: Perhaps the Best Understood Facet of the Problem

The foundational mathematical model and foundational disciplines of space robot mechanics were well articulated by Romano in 2020 [27], while Pantalone et. al, declared “providing a multi-function surgical robot is the new frontier.” [5]. Robot dynamics comprise the two constituent fields of kinetics and kinematics [28,29]. Kinetics describes the physics and mathematics of systems under the action of forces and torques, while kinematics describes mere geometric properties without application of forces and torques. These canonical definitions were elaborated specifically for robotics by Desoyer in the 1980s [30], where kinematics was highlighted as the “geometric relationships between robot coordinates”. Meanwhile, kinetics was highlighted as “…establishing the equations of motion…”, where Newton-Euler, Lagrange, and Gibbs-Appell equations were emphasized by Desoyer. (In Section 2.1.1 of this manuscript, Chasles’ Theorem is used instead of those listed by Desoyer). In the early 1990s, reference [31] sought to coin the phrases “forward kinetics” and “inverse kinetics”, blurring the distinction. These blurred definitions were largely not accepted, with the work cited only four times in thirty years. The focus of this present research resides in kinetics, while kinematics is assumed. Any equivalent geometric may be permissible to describe the joint angle configurations to access a point in Cartesian space. Since kinematics is arguably a (presently) dominant field of robotic studies, kinetics is more seldom studied. 

Thus, this manuscript focuses on kinetics beginning with seminal works and broadening to modern notions of Pontryagin’s methods embodied in systems theory, seeking to propose the most efficacious methods that are critically evaluated in direct comparisons.

#### 2.1.1. Kinetics (with Elaboration of Basic Material in Appendix C)

The current understanding of kinetics elaborates the equivalence of several forms of kinetic derivations described as first principles: Chasles’ theorem combining Newton’s law and Euler’s rotational moment equations, Hamilton’s principle, Lagrange’s equations, D’Alembert’s principle, and Kane’s equations (sometimes referred to as Lagrange’s form of D’Alembert’s principle). These expressions of kinetics (Chasles’ expression in particular) are labeled as constraints that must be obeyed in formulation of an optimization problem in systems theory. 

**Chasles’ theorem.** The most general rigid body displacement can be produced by a translation along a line (called its screw axis or Mozzi axis) followed (or preceded) by a rotation about an axis parallel to that line.[23,32]

#### 2.1.2. Evolution of Applying Basic Mechanics through Systems Theory

This manuscript applies modern systems theory to classical kinetics primarily expressed by Chasles via combination of Euler’s moment equations and Newton’s law for rotational and translation motion, respectively, as revealed in Equation (1). 

Kinematic elaboration is restricted to expression of motion in coordinates of rotating reference frames, resulting in what are sometimes called “fictional forces” arising from inclusion of the coupling, cross-products of the transport theorem in Equation (2) the left-side of which articulates translational instrument motion while the right-side articulates rotational instrument motion. The “named” terms (e.g., Euler, Coriolis, Centrifugal) are the so-called “frictional forces that arise from expressing motion in coordinates of rotating reference frames”.
sensors-23-01510-t002_Table 2Table 2Proximal variable definitions ^1^.VariableDefinitionVariableDefinitionFExternally applied forcesTExternally applied torquesmMassJMass moment of inertiaaTranslational accelerationαRotational accelerationx¨Translational accelerationθ¨Rotational accelerationd2xdt2Translational accelerationd2θdt2Rotational accelerationrRadius vector relative to rotating frameω˙Rotational accelerationvVelocity vector relative to rotating frame

^1^ Such tables are distributed throughout the manuscript to increase the ease of reading, while a combined, master table of definitions is included in the appendices.
(1)$$F=\mathrm{ma}=m\ddot{x}=m\frac{d^{2}x}{{\mathrm{dt}}^{2}}↔\;T=J\alpha =J\dot{\omega }=J\ddot{\theta }=J\frac{d^{2}\theta }{{\mathrm{dt}}^{2}}$$

**Transport theorem.** A vector equation that relates the time derivative of a Euclidean vector as evaluated in a non-rotating coordinate system may have its time derivative measured in a rotating reference frame.[33]


(2)
$$F=\mathrm{ma}+\underset{\mathrm{Euler}}{\underbrace{m\frac{d\omega }{\mathrm{dt}}\times r}}+\underset{\mathrm{Coriolis}}{\underbrace{2m\omega \times v}}+\underset{\mathrm{Centrifigual}}{\underbrace{m\omega \times \omega \times r}}↔\;T=J\dot{\omega }+\omega \times J\omega$$


### 2.2. Classical Benchmarks

The robotic space surgical system evaluated here is assumed to have been purchased with noisy sensors of angular position and velocity and a robust classical feedback controller. The commercially available da Vinci system does have direct haptic feedback capabilities, but the magnitude of force fed back to the surgeon during typical surgical maneuvers is so low as to have negligible effect on perception and performance [34]. The assumed controller described in Equation (3) has been illustrated to have zero steady-state error while having no integrator (avoiding integrator windup). Proportional plus velocity control utilizes proportional control by forming a state error scaled by a proportional gain adding a negative gained value of velocity (translational or rotational) as elaborated in Equation (3). The velocity channel is not a differentiated version of the position or angle channel, as is the case with classical cascaded control topologies of PD, PI, PID types (proportional plus derivative, proportional plus integral, and proportional plus integral plus derivative, respectively). 

**Final value theorem.** Mathematically, if function f(t) in continuous time with unilateral Laplace transform F(s), then the final value theorem establishes conditions under which $f\left(\infty \right)=\underset{s\to 0}{\lim}s\left(\frac{\theta \left(s\right)}{\theta_{d}\left(s\right)}\right)$.[35]

Assuming idealized governing differential equations (Equation (1)), the assumed control and closed loop system equations are provided in Equation (3), where the application of the final value theorem provides an expectation of zero steady-state error having systemically avoided integrator windup.
(3)$$\begin{array}{c} J\ddot{\theta }+K_{V}\dot{\theta }+K_{P}\theta =K_{P}\theta_{d}↔u=K_{p}\left(\theta_{d}-\theta \right)-K_{v}\dot{\theta }\\ \frac{\theta (s)}{\theta_{d}\left(s\right)}=\frac{K_{p}}{Js^{2}+K_{v}s+K_{p}}\to \;C.\;E.:\;s^{2}+K_{v}s+{\left.K_{p}\right|}_{I=1}\equiv s^{2}+2\xi \omega_{n}s+\omega_{n}^{2}\\ f\left(\infty \right)=\lim_{s\to 0}\;s\left(\frac{\theta (s)}{\theta_{d}\left(s\right)}\right)=0 \end{array}$$

**Table 3 sensors-23-01510-t003:** Proximal variable definitions ^1^.

Variable	Definition	Variable	Definition
$\theta_{d}$	Desired state trajectory	ξ	Critical damping ratio
$K_{P}$	Proportional gain	$\omega_{n}$	Natural frequency
$K_{v}$	Velocity gain	$\omega_{b}$	Controller bandwidth
$t_{s}$	Settling time	$t_{r}$	Rise time

^1^ Such tables are distributed throughout the manuscript to increase the ease of reading, while a combined, master table of definitions is included in the appendices.

Equations (1) and (2), respectively, were controlled by Equation (3), and the *goal is to impose improved performance on the purchased control system* (not being able to replace the purchased electronics with a preferred system). Assuming Equations (1)–(3) are provided in the purchased system, gains were permissibly tuned for performance specification by equating the ubiquitous closed-loop system Equation (3) to the performance specified. The desired rise time (an avatar for rapidity of instrument movement) established the system natural frequency per $t_{r}=\frac{1.8}{\omega_{n}}$ where $\omega_{n}\approx \omega_{b}$ is the desired control bandwidth, therefore $\omega_{n}=\frac{1.8}{t_{r}}\to K_{p}=\omega_{n}{}^{2}$. Settling time as an avatar for system precision was incorporated into the design to limit oscillation to stabilize within 2–5% percent of steady state, establishing the closed-loop system’s damping ration and thus velocity gain term: $t_{s}=\frac{4.6}{{\xi \omega }_{n}}$ $\to \xi =\frac{4.6}{t_{s}\omega_{n}}\to K_{v}=2{\xi \omega }_{n}$. Elimination of differentiation in the derivative channel (utilized by classical proportional plus derivative, PD methods…r) often bestows relative advantage tracking desired trajectories, while also eliminating the necessity of integral control inclusion. 

### 2.3. Induce Improvement: Systems Theory, Prefiltering, Decoupling Nonlinear Transport Theorem

Systems theory is a modern approach to assembling mathematical necessary conditions of optimality and using their application to reduce the problem of controlling the robotic instrument to a boundary-value problem of several, perhaps many, differential equations [16]. Real-time implementation of the results can be exacerbated by the necessity of inverting poorly conditioned (sometimes rank-deficient) matrices [21,22,23]. Meanwhile, transport theorem may be decoupled in the control design by adopting the physics-based methods of [17]. The following sections elaborate utilization of systems theory, real-time implementation (utilizing sensors), pre-filtering (proposed to induce performance on the pre-existing commercial control system), and finally, expansion of designs to account for nonlinear, coupling transport theorem.

#### 2.3.1. Open Loop Optimization Via Systems Theory (A Theoretical Comparative Benchmark)

Minimize Equation (4) constrained by the dynamic constraints in Equation (1) where the system is assumed to start at rest and end at an angular position scaled by its final value such that the final scaled angular position is unity. Utilizing three-dimensions of rotational motion yields six equations through augmentation with the dual dynamics embodied in the adjoint equations. The boundary value problem is established by these six equations necessitating six boundary values, where co-state boundary values may be provided by the terminal transversality condition of the endpoint Lagrangian. This procedure is well articulated identically in [16,17,21,22,23] whose results are Equations (5) and (6) for a point to point maneuver of the robotic medical instrument.
(4)$$\mathrm{Cos}t\;\left(\mathrm{effort}\right)=\frac{1}{2}\overset{\infty }{\underset{0}{\int }}\left(u^{T}u\right)\mathrm{dt}$$
(5)$$u^{*}=\mathrm{at}+b,\;{\ddot{\theta }}^{*}=\mathrm{at}+b,\;{\dot{\theta }}^{*}=\frac{1}{2}{\mathrm{at}}^{2}+\mathrm{bt}+c,\;\theta^{*}=\frac{1}{6}{\mathrm{at}}^{3}+\frac{1}{2}{\mathrm{bt}}^{2}+\mathrm{ct}+d$$
(6)$$u^{*}=-12t+6,\;{\ddot{\theta }}^{*}=-12t+6,\;{\dot{\theta }}^{*}=-6t^{2}+6t,\;\theta^{*}=-2t^{3}+3t^{2}$$

**Table 4 sensors-23-01510-t004:** Proximal variable definitions ^1^.

Variable	Definition	Variable	Definition
$u,{\;u}^{*}$	Control, minimum control	${\ddot{\theta }}^{*}$	Control-minimizing acceleration
a,b,c,d	Constants of integration	${\dot{\theta }}^{*}$	Control-minimizing velocity
t	Time (dimensionless)	$\theta^{*}$	Control-minimizing position

^1^ Such tables are distributed throughout the manuscript to increase the ease of reading, while a combined, master table of definitions is included in the appendices.

Here, the assertion made in the description of literature gaps and innovations in Section 1.1 is partially manifest. Equation (6) reveals an instrument control signal that solves the constrained optimization program, minimizing the effort of articulating the medical instrument, but the equation contains much more, and that additional content is useful to impose improvements on inaccessible inner electronics of commercial medical space robots. The instrument trajectories that minimize pointing effort are also revealed, whereby simple differentiation and integration, any motion state, may be discerned: position, integral of position, velocity, acceleration, jerk, etc. Section 2.3.4 will take advantage of this feature to decouple the nonlinear, coupling effects of transport theorem using the optimal angular velocity profiles in Equations (5) and (6), where Equation (6) embodies the open-loop solution of a point-to-point maneuver from quiescent initial condition, while Equation (5) is the form used for real-time optimization from non-quiescent initial condition, using (poor, noisy, fused) sensor data. 

Rather than adopt the optimal control and seek to replace the inaccessible electronics of the commercial medical space robot, consider adopting a fuller extend of the solutions expressed in Equations (5) and (6), and instead seek to use that fuller extend to impose improvements.

#### 2.3.2. Real-Time Optimization (and Singular Switching)

The previous section of text expressed that Equation (5) could be useful amidst poor, fused noisy sensors to formulate real-time effort minimization from non-quiescent initial conditions. Using the angle and angular rate equations from Equation (5) expressed in Equation (7), the initial and final angular position and rate may be written as a matrix-vector system of equations. Next, the initial and final attitude and rate may be replaced with the current attitude $\theta \left(t\right)$ and rate $\dot{\theta }\left(t\right)$ obtained by poor, fused noisy sensors permitting real-time solution of the coefficients (a,b,c,d) by inverting the matrix in Equation (8) producing estimates ($\hat{a},\hat{b},\hat{c},\hat{d}$).
(7)$${\dot{\theta }}^{*}=\frac{1}{2}{\mathrm{at}}^{2}+\mathrm{bt}+c,\;\theta^{*}=\frac{1}{6}{\mathrm{at}}^{3}+\frac{1}{2}{\mathrm{bt}}^{2}\;\&\;{\ddot{\theta }}^{*}\left(t_{f}\right)=\frac{1}{2}a+b=0,\;\theta^{*}\left(t_{f}\right)=\frac{1}{6}a+\frac{1}{2}b=1$$
(8)$$\underset{\left[T\right]}{\underbrace{\left[\begin{array}{cccc} \frac{t_{0}^{2}}{6} & \frac{t_{0}^{2}}{2} & t_{0} & 1\\ \frac{t_{0}^{2}}{2} & t_{0} & 1 & 0\\ \frac{1}{6} & \frac{1}{2} & 1 & 1\\ \frac{1}{2} & 1 & 1 & 0 \end{array}\right]}}\underset{\left\{p\right\}}{\underbrace{\left\{\begin{array}{c} a\\ b\\ c \end{array}\right\}}}=\underset{\left\{b\right\}}{\underbrace{\left\{\begin{array}{c} \theta_{0}\\ {\dot{\theta }}_{0}\\ 1\\ 0 \end{array}\right\}}}\to \left\{\begin{array}{c} \hat{a}\\ \hat{b}\\ \hat{c}\\ \hat{d} \end{array}\right\}={\left[\begin{array}{cccc} \frac{t_{\mathrm{now}\;}^{2}}{6} & \frac{t_{\mathrm{now}\;}^{2}}{2} & t_{0} & 1\\ \frac{t_{\mathrm{now}\;}^{2}}{2} & t_{\mathrm{now}\;} & 1 & 0\\ \frac{1}{6} & \frac{1}{2} & 1 & 1\\ \frac{1}{2} & 1 & 1 & 0 \end{array}\right]}^{-1}\left\{\begin{array}{c} \theta (t)\\ \dot{\theta }\left(t\right)\\ 1\\ 0 \end{array}\right\}\&u^{*}\equiv \hat{a}t+\hat{b}$$

To ensure values of (a,b,c,d) are available for later use (e.g., utilizing effort-minimizing trajectories and controls), some attention must be given to the necessary inversion of the square $\left[T\right]$ matrix in Equation (8), whose dimension is [4 × 4]. As scaled time propagates from zero to unity, the initial value $t_{0}$ is replaced with $t_{\mathrm{now}}$, and the initial instrument angular position and rate are replaced with the current values, permitting the present values to be used in the optimization problem formulation as initial values. As scaled time approaches unity, the columns and rows of the matrix $\left[T\right]$ become linearly dependent, causing the matrix to lose rank, making inversion problematic. Several techniques are possible to monitor the matrix condition (e.g., the matrix determinant, the matrix condition number, etc.). Singular switching is proposed. Based on the inverse of the matrix condition number, the switch either utilizes the well-conditioned matrix inverse values to substantiate the effort-minimizing values of ($\hat{a},\hat{b},\hat{c},\hat{d}$) or alternatively uses the open-loop values (a,b,c,d). Fortunately, the matrix remains generally well-conditioned although switching is strictly necessary in the final few time-steps as the robot instrument approaches the final orientation. 

With the understanding that real-time estimates of effort-minimizing trajectories are available for (later) decoupling the nonlinear transport theorem, next consider how to seemingly negate the signals generated by the pre-existing electronics. Inspired by “system inversion” [36,37], consider options for prefiltering the surgeon’s commands generated by the pre-existing systems accepting the surgeon’s inputs (e.g., as depicted in Figure 4). 

#### 2.3.3. Inducing Improvements: Controlled Plant Inverting Prefilter

Seeking to patch over the signals from the existing electronics, notice the expression of those electronics in Equation (3) embody linear, time-invariant equations expressed in so-called Laplace domain or “s” domain leading to an output/input relationship called a transfer function. The inverse of that transfer function is an input/output relationship that is useful to pre-filter the surgeon’s inputs such that the signal reaching the commercial robotic instrument contains the inverse of the treatment anticipated by the inaccessible electronics of that instrument. At least two options are available: (1) seek to negate the dominant open-loop dynamics embodied in the so-called double-integrator or (2) seek to negate the entire closed-loop response of the electronics. Both pre-filtering topologies are illustrated in the appendix to aid replication of the work. Assuming the viability of one or both approaches permits imposition of improved responses utilizing the effort-minimizing trajectories of Section 2.3.2 to decouple the deleterious effects of transport theorem. 

The input to the feedback system is a position, while the optimal control is acceleration. The nature of the input must be maintained, so the operating physician can continue to simply command their desired position to the medical instrument, as depicted in Figure 4. The physician gives inputs to the pre-existing control interface, while the prefilter uses the results of Section 2.3.2 and Section 2.3.3 to establish effort-minimizing trajectories that are prefiltered and sent to the pre-existing electronics, where the pre-filtering is designed to negate the efforts of the pre-existing system’s signals in favor of imposing improved performance.
(9)$${\dot{\theta }}^{*}=\frac{1}{2}{\mathrm{at}}^{2}+\mathrm{bt}+c,\;\theta^{*}=\frac{1}{6}{\mathrm{at}}^{3}+\frac{1}{2}{\mathrm{bt}}^{2}\;\&\;{\ddot{\theta }}^{*}\left(t_{f}\right)=\frac{1}{2}\mathrm{at}+b=0,\;\theta^{*}\left(t_{f}\right)=\frac{1}{6}a+\frac{1}{2}b=1$$
(10)$$\tau \equiv u=K_{p}\left(\theta_{d}-\;\theta \right)-K_{v}\to \theta_{d}=\frac{u}{K_{p}}+\theta +\frac{K_{v}}{K_{p}}\dot{\theta }$$
(11)$$\mathrm{Defining}\;\tau^{*}\equiv u^{*}\to \theta_{d}^{*}\left(t\right)=\frac{u*}{K_{p}}+\theta^{*}\left(t\right)+\frac{K_{v}}{K_{p}}{\dot{\theta }}^{*}\left(t\right)\;\;\forall t\in \left[t_{0},t_{f}\right]$$
(12)$$\theta_{d}^{*}\left(t\right)=\frac{1}{K_{p}}\left(u^{*}\left(t\right)+K_{v}\int u^{*}\left(t\right)\mathrm{dt}+K_{p}\int \int \left(u^{*}\left(t\right)\mathrm{dt}\right)\mathrm{dt}\right)$$
(13)$$\Theta_{d}^{*}\left(s\right)=\frac{1}{K_{p}}\left(U^{*}\left(s\right)+\frac{K_{v}}{s}U^{*}\left(s\right)+\frac{K_{p}}{s^{2}}U^{*}\left(s\right)\right)$$
(14)$$\Theta_{d}^{*}\left(s\right)=\left[\frac{s^{2}+K_{v}s+K_{p}}{K_{p}s^{2}}\right]U^{*}\left(s\right)$$
(15)$$\frac{\Theta_{d}^{*}\left(s\right)}{U^{*}\left(s\right)}=\frac{1}{K_{p}}\left[\frac{s^{2}+K_{v}s+K_{p}}{s^{2}}\right]$$

**Table 5 sensors-23-01510-t005:** Proximal variable definitions ^1^.

Variable	Definition	Variable	Definition
θ	Medical instrument rotation angle	a,b,c,d	Effort-minimizing constants
$\theta_{d}$	Desired rotation angle	dt	Differential scaled time
$\theta^{*}$	Effort-minimizing rotation angle	t	Scaled time
$\dot{\theta }$	Medical instrument angular rate	$t_{0}$	Initial scaled time
${\dot{\theta }}^{*}$	Effort-minimizing angular rate	$t_{f}$	Final scaled time
$\Theta_{d}^{*}$	Desired rotation angle (Laplace domain)	$U\left(s\right)$	Effort (Laplace domain)
$K_{v}$	Velocity gain	$\tau^{*}\equiv u^{*}$	Effort-minimized torque
$K_{p}$	Proportional gain	$U^{*}$	Minimized effort (Laplace domain)

^1^ Such tables are distributed throughout the manuscript to increase the ease of reading, while a combined, master table of definitions is included in the appendices.

Equation (9) instantiates the utility of the trajectories derived by systems theory in Section 2.3.2 Substituting these effort-minimizing relationships into the purchased control system, Equation (10) yields the relationships in Equations (11) and (12) between the desired (effort-minimized) medical instrument attitude angle and the purchased systems’ indigenous electronics. Leaving the time-varying, effort-minimizing results in variable form as $u^{*}$ reveals the desired output Equation (13), which is subsequently reformulated into an output/input transfer function through Equations (14) and (15). 

Notice the utilization of the effort-minimizing control from systems theory thus far, but there is an additional opportunity to utilize other aspects of the effort-minimizing solutions. One such opportunity proposed is utilization of the effort-minimizing angular rate profiles derived from systems theory to decouple the deleterious, nonlinear coupling transport theorem highlighted in Equation (2). 

#### 2.3.4. Decoupling Nonlinear Transport Theorem

From Equation (2) is $T=J\dot{\omega }+\omega \times J\omega$ where resultant torque T in the canonical equation is represented by commonplace variable τ in later equations this manuscript in accordance with the canonical practices of each discipline (e.g., dynamics, optimization, autonomy, controls, etc.) and angular rate ω and acceleration $\dot{\omega }$ are represented as in Equation (1) as $\ddot{\theta }$ and $\dot{\theta }$, respectively. Thus, the effort minimizing ${\dot{\theta }}^{*}$ and $\ddot{\theta }$ from Equation (9) are combinable in the fashion of Equation (2) to articulate an effort-minimized differential equation that includes the nature of nonlinear, coupling transport theorem as expressed in Equation (16). Substituting the values derived from systems theory for the effort-minimizing trajectories from Equation (9) into Equation (16) produces the real-time version akin Section 2.3.2, where decoupling is feedforward. Alternatively, feedback of acceleration and velocity states permits noisy sensor signals to be recombined for feedback decoupling. Singular switching is implemented identically as in Section 2.3.2 as the effort-minimizing values of (a,b,c,d) become resonant.
(16)$$T=J\dot{\omega }+\omega \times J\omega \;\to {\;\tau }^{*}=J{\ddot{\theta }}^{*}+{\dot{\theta }}^{*}\times {J\dot{\theta }}^{*}$$

#### 2.3.5. Intermediate Summary

Section 2.3.1 illustrated how to utilize systems theory to formulate effort-minimizing controls in addition to trajectories for instrument angular position (and its integral), rate, acceleration, jerk, etc., for a given point-to-point maneuver, while Section 2.3.2 illustrated the manner of making the calculations real-time, utilizing the fusion of assumed low-quality, noisy sensors. Section 2.3.3 revealed a method to negate the signals from the pre-installed electronics on the commercially purchased robotic medical system, while Section 2.3.4 proposed a method for including the (oft neglected) nonlinear, coupling transport theorem. 

To aid replication of the research, pseudo-code for implementation is provided in the appendix using topologies from the graphic user interface of SIMULINK^®^.The simulation tests are described next in Section 2.4. 

### 2.4. Simulation Tests in SIMULINK^®^

Table 6 lists the dozen investigations performed. The first four articulated are idealized to help to establish baseline theoretical performances of idealized systems operating in idealized circumstances. Topologies and depictions of simulation systems are included in the Appendix A. 

#### 2.4.1. Method 1

Idealized double-integrator dynamics controlled by classical “velocity control”, sometimes labeled as P + V control, which is different than proportional + derivative or PD control. This method is articulated in Equation (3) as introduced by Banginwar in [39].

#### 2.4.2. Method 2

Idealized double-integrator dynamics controlled by modern open-loop optimal methods obtained using systems theory. This method is articulated in Equation (6) as proposed in [17].

#### 2.4.3. Method 3

Idealized double-integrator dynamics controlled by state-of-the-art real-time optimal control techniques. This method is articulated in Equation (8) as introduced by Banginwar in [39].

#### 2.4.4. Method 4

Idealized double-integrator dynamics controlled by state-of-the-art real-time optimal control techniques recently augmented with automatic singular switching. This method is also articulated in Equation (8), where special attention is urged to the inversion of the potentially rank-deficient matrix [T] as proposed by Banginwar in [39].

The subsequent eight methods in Table 6 (methods 8–12) are non-idealized systems operating in non-idealized circumstances, including randomly varying mass and mass moments and fusion of poor, noisy sensors. The first four (methods 5–8) also include nonlinear coupled transport theorem in the robotic system dynamics, but those deleterious inclusions are not specifically accounted for in the control design. Meanwhile, the final four techniques (methods 8–12) specifically augment the control design with the proposed transport theorem decoupling per Equation (16). Furthermore, the final four techniques include proposed pre-filtering. Topologies and depictions of simulation systems are included in the appendix.

Essentially, methods 8–12 implement all the proposed methods and should be directly compared to method 5, while all methods may be compared to idealizations represented in cases 1–4 merely out of mathematical interest.

#### 2.4.5. Method 5

Non-idealized double-integrator dynamics with nonlinear, coupling transport theorem controlled by a proposed pre-filtering of classical “velocity control” as described in Section 2.4.1.

#### 2.4.6. Method 6

Non-idealized double-integrator dynamics with nonlinear, coupling transport theorem controlled by a proposed pre-filtering of modern open loop optimal control derived using systems theory as described in Section 2.4.2.

#### 2.4.7. Method 7

Non-idealized double-integrator dynamics with nonlinear, coupling transport theorem controlled by a proposed pre-filtering of real-time optimal control as described in Section 2.4.3.

#### 2.4.8. Method 8

Non-idealized double-integrator dynamics with nonlinear, coupling transport theorem controlled by a proposed pre-filtering by state-of-the-art real-time optimal control techniques recently augmented with automatic singular switching as described in Section 2.4.4.

#### 2.4.9. Method 9

Non-idealized double-integrator dynamics with nonlinear, coupling transport theorem controlled by a proposed pre-filtering of classical “velocity control”.

#### 2.4.10. Method 10

Non-idealized double-integrator dynamics with nonlinear, coupling transport theorem controlled by a proposed pre-filtering of classical “velocity control”.

#### 2.4.11. Method 11

Non-idealized double-integrator dynamics with nonlinear, coupling transport theorem controlled by a proposed pre-filtering of real-time optimal control.

#### 2.4.12. Method 12

Non-idealized double-integrator dynamics with nonlinear, coupling transport theorem controlled by a proposed pre-filtering of real-time optimal control with singular switching.

Due to the high number of long labels describing the dozen approaches studies, Table 6 adopts a numbering of methods with shorter labels (e.g., “Method 1, etc.). 

The results of the dozen investigations in this study are presented next in Section 3, the Results. The presentation of results follows the structure of Table 6, so the reader can remind themselves of the various methods and what proposals are present in each method. The results are presented as raw data first using typical figures of merit, e.g., errors obeying the physicians commanded instrument angle and rate, cost (effort), and computation burden (represented by computer run-times). To permit direct comparison by the readers, cases are declared as comparative benchmarks, and the results are then presented as percent performance improvement compared to the benchmarks in Section 4, the Discussion. 

## 3. Results

Using numerical simulations whose SIMULINK^®^ topologies are included in the appendix based on the methods presented in Section 2, this section presents comparisons of one dozen disparate approaches and makes recommendations for further development based on commonplace figures of merit (e.g., attitude and rate errors, cost (effort), and computational burden). Table 7 and Table 8 contain figures of merit corresponding to the data plotted in Figure 5.

### Instrument Pointing Accuracy, Cost (Effort), and Computational Burdens

Commanding a simple, one-dimensional maneuver facilitates comparison using ubiquitous figures of merit (e.g., final attitude and rate errors, cost (effort), and computational runtime). Having executed the identical instrument maneuver in all cases, Table 7 and Figure 5 reveal the results of the eight disparate approaches (methods 5–12), while the first four cases (methods 1–4) were mathematical idealizations with double-integrators illustrating essences of each method, e.g., methods 2 and 4 open loop optimal and real-time optimal, respectively, achieve mathematically minimal costs and benchmark attitude and rate errors in a fictionally idealized world with no transport theorem. 

**Figure 5 sensors-23-01510-f005:**
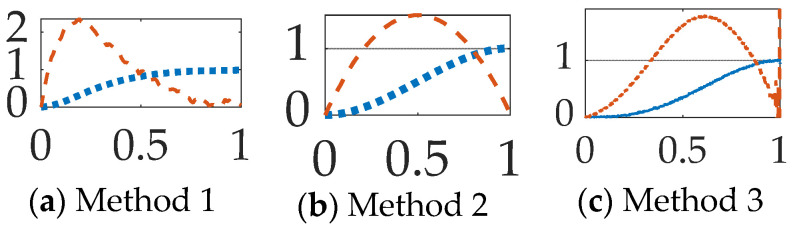
Simulation test results: subfigures (**a**–**l**) display qualitative results, while corresponding quantitative figures of merit are in Table 7 inserted into subfigure. The abscissa displays scaled time normalized to vary between zero and unity. The ordinant displays displacement and rate, respectively represented by dotted and dashed lines.

**Table 7 sensors-23-01510-t007:** Quantitative figures of merit corresponding to the qualitative data presentation in accompanying Figure 5a–l ^1^.

Case ^1^	$\theta \left(t_{f}\right)$	$\omega \left(t_{f}\right)$	Runtime	Cost, J
Idealized cases (for theoretical comparison)				
Method 1	0.9812	0.0464	2.25	35.2
**Method 2**	**1.0093**	**0.0054**	**2.12**	**6**
Method 3	1.1836	46.6302	2.25	333,557
Method 4	1.0093	0.0053	2.21	6
Realistic cases (Compare to Methods 1–4)				
**Method 5**	**0.9596**	**0.0595**	**2.23**	**16.1**
Method 6	0.1831	−1.071	2.24	3.6
Method 7	1.2004	0.8998	2.24	360,604
Method 8	0.1831	−1.7071	2.21	3.6
Method 9	0.9917	−0.0267	2.27	16.2
**Method 10**	**0.9917**	**−0.0267**	**2.19**	**16.1**
Method 11	1.1997	1.0373	2.21	388,346
Method 12	0.6198	−0.2051	2.18	3.6

^1^ Cases using methods 1–4 are idealized to establish mathematically best possible performance. Methods 5–12 include randomly varied mass and fusion of noisy state and rate sensors. Methods definitions are in Table 6 and correspond to Figure 5.

Figure 5a displays the performance of method 1, idealized double-integrator dynamics controlled by classical “velocity control”, sometimes labeled as P + V control, which is different from proportional + derivative or PD control. This method is akin to a method used in a college course, and the performance should be familiar, where proportional gain terms permit relatively steep rise time, while the velocity gain prevents overshoot, while the expended effort was relatively high. Figure 5b displays the slightly more complicated case where the same idealized dynamics in Figure 5a are controlled using modern open-loop optimal methods obtained using systems theory. The results appear perfectly symmetric (lacking high rise time and gradual settling), while the effort is mathematically minimized. Figure 5c illustrates performance of real-time optimal control techniques on the same idealized dynamics, while Figure 5d displays the efficacy of augmentation with automatic singular switching. Performance lacking singular switching suffers degradation caused by inversion of a poorly conditioned matrix, while implementation of singular switching restores the appearance of optimum results akin the idealized open loop case (matching both tracking performance and control effort minimization). 


*General conclusion of the idealized cases: feedback control is implementable with singular switching providing effort minimization with superlative tracking performance.*


These first four subplots reveal idealized performances, while the subsequent eight subplots reveal the efficacies of realistic cases. Figure 5e displays the high efficacy of the proposed pre-filtering of classical “velocity control” to control realistic, non-idealized double-integrator dynamics with nonlinear, coupling transport theorem. The accompanying figures of merit displayed in Table 7 indicate the method has the best position (or angle) tracking performance, the second-best rate tracking performance and lowest effort. The proposed pre-filtering of modern open loop optimal control derived using systems theory performed poorly, as displayed in Figure 5f, lacking feedback. Inclusion of feedback, the proposed pre-filtering of real-time optimal control derived using systems theory also performed poorly as displayed in Figure 5g. Augmentation with automatic singular switching did not greatly improve performance, as evidenced by Figure 5h; meanwhile, superlative performance was achieved by the proposed prefiltering of classical “velocity control”.


*General conclusion of the realistic cases: Proposed pre-filtering induces performance of a pre-existing classical control scheme, achieving superlative tracking performance and low effort, where the input output relationship is displayed in Figure 6. Furthermore, prefiltered open-loop optimal control augmented with transport theorem decoupling performed similarly well.*


## 4. Discussion

Recent innovations in systems theory promise effort-minimizing instrument control, but commercial medical robotics have pre-existing electronics, necessitating methods to “induce” effort-minimizing performance upon the pre-existing electronics omnipresence in the seminal literature. The literature review included classical control, optimal control, nonlinear optimal control, nonlinear optimal control, and learning control in addition to complementary features, such as autonomous trajectories and deterministic artificial intelligence. This manuscript focuses on space robot kinetics, broadening to modern notions Pontryagin’s methods embodied in systems theory, seeking to propose the most efficacious methods that are critically evaluated in direct comparisons. A dozen disparate approaches were implemented in direct comparison, and canonical figures of merit revealed superior performing methods. 

Based on a combination of position and rate errors, computational burden and effort, the results indicate the best choice for inducing changes in commercially purchased robotic space instruments is the negation of the effects of the purchased systems’ electronics using the proposed prefiltering, plus open-loop optimal control commands augmented with transport theorem decoupling using the proposed nonlinear combination of the effort-minimizing rate trajectories. Combined, the proposed methods lead to one percent and three percent performance improvement, respectively, for instrument angular positioning and rate with minimal effort and a two percent performance improvement in computational burden, while utilizing the fusion of poor, noisy sensors and assuming randomly variations in system parameters. The implication is that remote physicians have new options for increasing robust and precise performance of medical instruments in space. 

It is noteworthy to highlight the surprisingly degraded performance of real-time optimal implementations with singular switching when the prefilters and transport theorem decoupling were added (relative to the better performance in the cited literature). The poor performance compared to the cited literature inspires direction for future research. 

### Future Research

The results should next be validated in real practice, including comparisons to this manuscript and the cited literature. Sequel research is already funded to investigate the augmentation of the rigid-body treatments in this study with multi-body dynamics and accompanying control (described earlier as using the finite-element method together with nonlinear transport theorem). The effects of sudden changes in mass inertia properties and disturbances will be addressed as well. 

**Table 8 sensors-23-01510-t008:** *Performance comparison:* instrument attitude and rates indicate error percentages, while runtime and cost displays are percent performance improvements over comparative benchmark cases 2 and 5, respectively. Methods 1–4 are compared to idealized method 2 (lacking transport theorem); while cases 6–12 are compared to method 10 realistic cases with transport theorem ^1^.

	Case ^1^	$\theta \left(t_{f}\right)$	$\omega \left(t_{f}\right)$	Runtime	Cost, J
**Prefiltered P + V**	**Method 5**	**4%**	**6%**	**--**	**--**
Prefiltered Open-loop optimal	Method 6	82%	107%	0%	−78%
Prefiltered RTOC	Method 7	20%	90%	0%	2,242,839%
Prefiltered switched RTOC	Method 8	82%	171%	−1%	−78%
Prefiltered P + V + transport decoupling	Method 9	1%	3%	2%	1%
**Prefiltered Open-loop optimal + transport decoupling**	**Method 10**	**1%**	**3%**	**−2%**	**0%**
Prefiltered RTOC + transport decoupling	Method 11	20%	104%	−1%	2,415,393%
Prefiltered switched RTOC + transport decoupling	Method 12	38%	21%	−2%	−78%

^1^ All cases include randomly varied mass and fusion of noisy state and rate sensors. Methods correspond to definitions in Table 6.

## Figures and Tables

**Figure 2 sensors-23-01510-f002:**
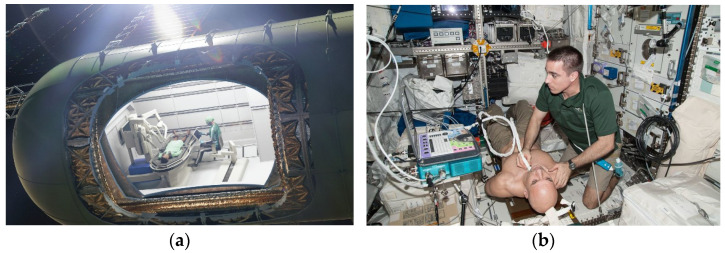
(**a**) Virtually rendered cutaway view of a postulated trauma pod surgical module with a four-armed surgical robot in the module. The patient is tethered to the operating table, while the assistant, using a touchscreen console, is tethered to the module structure via a movable chair. Illustration by T. Trapp (CC BY-SA 4.0). [6] (**b**) In the International Space Station’s Columbus laboratory, NASA astronaut Chris Cassidy performs an ultrasound on ESA astronaut Luca Parmitano for the Spinal Ultrasound investigation. The Spinal Ultrasound investigation sought to understand astronaut height increases while advancing medical imaging technology, by testing a smaller and more-portable ultrasound device aboard the station. Photo taken from [7] in compliance with NASA’s image use policy [8].

**Figure 3 sensors-23-01510-f003:**
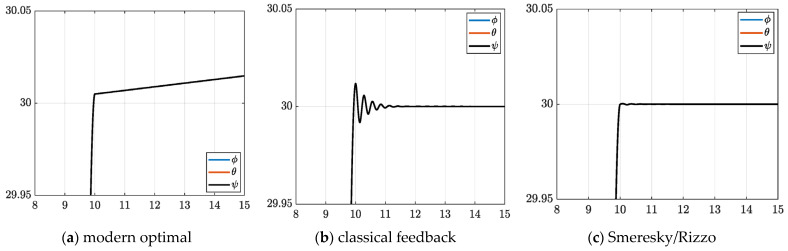
The results of Smeresky and Rizzo [20]: rotation degrees vs. time [seconds]. (**a**) results achieved using modern optimal methods; (**b**) results achieved using contemporary classical methods. (**c**) Smeresky’s and Rizzo’s [18] results achieved using their proposed methods. Comparisons such as this are deemed valuable, and this general methodology is adopted in this manuscript as a starting point.

**Figure 4 sensors-23-01510-f004:**
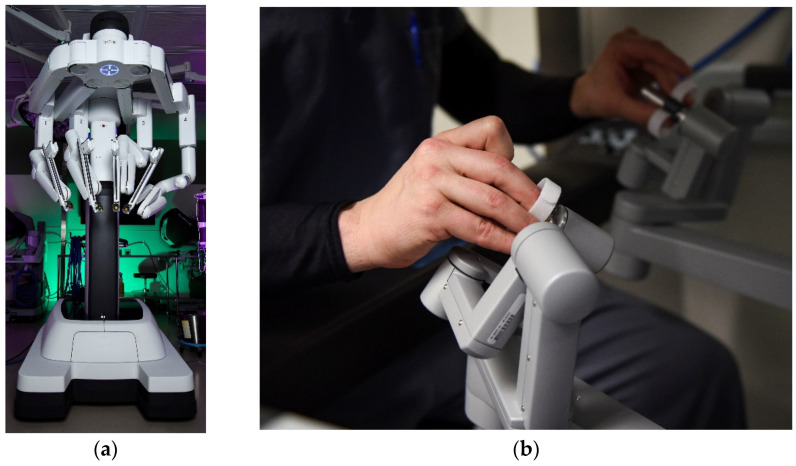
The 99th Surgical Operations Squadron (MSGS) performed their first robotic general surgery using the da Vinci surgery system, April 3, 2019, at the Mike O’Callaghan Military Medical Center. The depicted da Vinci robotic surgery system enables surgeons to perform complex procedures with precision and accuracy. (**a**) Shriever Space Force Base da Vinci surgery system [38]. (**b**) General surgeon provides inputs to command the operation of the da Vinci surgery system [38]. The system must accept these commands and provide precision instrument pointing to ensure patient safety.

**Figure 6 sensors-23-01510-f006:**
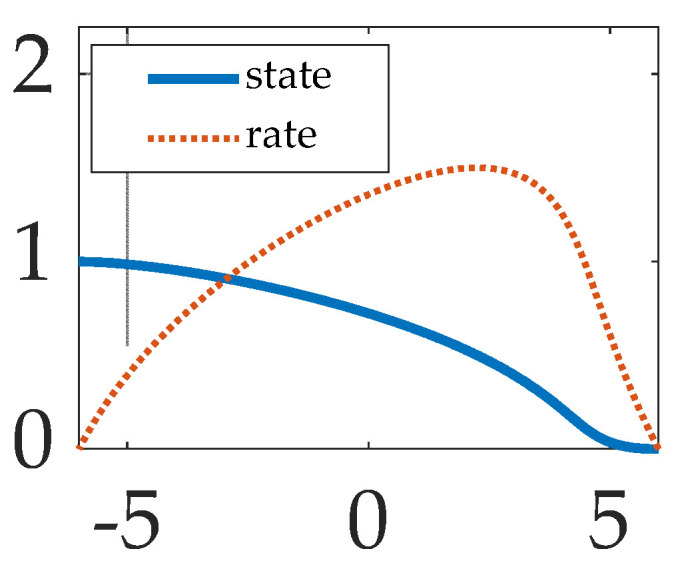
Input output relationship of the proposed pre-filtering technique. The abscissa is the control, while the ordinate is the state and rate trajectories, respectively.

**Table 6 sensors-23-01510-t006:** Brief numbering of the dozen approaches studied. Short case labels (e.g., “Method 1, etc.) are used throughout Section 2 and Section 3 before finally being abandoned for the sake of clarity in the presentation of short-listed results in Section 4 ^1^.

Figure	Case ^1^	Plant	Control
2a	Method 1	Double integrator (idealization)	P + V
2b	Method 2	Double integrator (idealization)	Open-loop optimal
2c	Method 3	Double integrator (idealization)	RTOC
2d	Method 4	Double integrator (idealization)	Switched RTOC
2e	**Method 5**	**Double integrator, transport theorem**	**Prefiltered P + V**
2f	Method 6	Double integrator, transport theorem	Prefiltered Open-loop optimal
2g	Method 7	Double integrator, transport theorem	Prefiltered RTOC
2h	Method 8	Double integrator, transport theorem	Prefiltered switched RTOC
2i	Method 9	Double integrator, transport theorem	Prefiltered P + V + transport decoupling
2j	**Method 10**	**Double integrator, transport theorem**	**Prefiltered Open-loop optimal + transport decoupling**
2k	Method 11	Double integrator, transport theorem	Prefiltered RTOC + transport decoupling
2l	Method 12	Double integrator, transport theorem	Prefiltered switched RTOC + transport decoupling

^1^ All cases include randomly varied mass and fusion of noisy state and rate sensors. Methods correspond to Table 7 and Table 8.

## Data Availability

Details regarding where data supporting reported results can be obtained by contacting the corresponding author.

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
