# Peer review of "Inducing Performance of Commercial Surgical Robots in Space"

_sensors, 2023, doi:10.3390/s23031510_

Round 1

Reviewer 1 Report

Reviewer’s report on a paper submitted to the journal of Sensors, with the tile of “Inducing performance of commercial surgical robots in space

The topic of the paper is interesting. However, there are several drawbacks which must be addressed before the paper can be considered further.

The paper abstract doesn’t represent the main contribution of the work. What’s new in this work? It should be better highlighted. In particular, Abstract should be re-written. Abstract should outline the key novelties of the research and briefly describe the findings/achievements.

There are some papers reviewed in subsections 1.1 and 1.2; however, the literature review is not complete. The author(s) should review the state-of-the-art and report some of the challenges in the field of surgical robots. The numbering for the subsection “1.1 Literature gaps and innovations proposed” should also be amended to “1.3”.

Section 2 is so lengthy and there are, in some cases, pretty basic materials there. It can be shortened by removing some content and giving references for readers to refer to.

Sections 3 and 4 must be expanded. The results and discussion parts are too short. The author(s) need to develop some further discussions about how the results can be validated in real practice? Any comparisons with previous studies published in the literature?

Conclusion and future directions should be expanded too. What were the main findings of this research?

Author Response

  1. Abstract doesn’t represent the main contribution of the work. What’s new in this work? It should be highlighted. In particular, abstract should be re-written to outline key novelties and findings/achievements.
  • Abstract has been improved to specifically highlight the new work and novel findings/achievements. Thanks for the revision recommendation.
  1. Literature review is not complete. Authors should review the state of the art and report some challenges in the field. Numbering should be amended.
  • State of the art and challenges in the field are highlighted in the review and numbering has been amended. Thanks for the revision recommendation.
  1. Section 2 can be shortened removing basic content in favor of references.
  • Appendix C is newly created to host the basic content.
  1. Sections 3 and 4 must be expanded. Results and discussion parts are too short. Authors need to develop further discussions about how the results can be validated in real practice including comparisons with previous studies in the literature.
  • The sections are modified accordingly, thanks.
  1. Conclusions and future directions should be expanded. What were the main findings of the research?
  • Expansion has been added to the Results section providing stronger tie to the Discussion.

Reviewer 2 Report

The work presented in the paper entitled ‘Inducing performance of commercial surgical robots in space’ is interesting. Resolving the following comments will help the author to improve his/her presented work.

1.         The literature presented in the Subsection 1.1 (Medical procedure free-floating in space) is very generic. It would be nice if the author present that what were the outcomes of performing the diagnostics/surgical procedure. Moreover, it would be grate if the author highlights the encountered challenges in the diagnostics /surgical method along with the advantages and disadvantages of the methods.

2.         Please correct the spelling of ‘hihgly’ in Subsection 1.2.

3.         In the Subsection 1.2, it would be great if the author can explain what the major challenges in the manipulation are, particularly for the space surgical applications.

4.         Based on the challenges which author would discuss in Subsection 1.2, it would be important to highlight that how these challenges were addressed in the past and what are the gaps in the technology.

5.         Referring to the line no. 104 to 117, the author is discussing some trajectory tracking or velocity based conventional controls methods for manipulation. According to my opinion there are several advanced control methods that have been implemented to solve the complex manipulation problem.

a.         For the reader it would be interesting to know that what would be the affect of mass inertia properties on the manipulation in the Space environment.

b.         What kind of disturbances and uncertainties a manipulator can experience while working in the space. Also, I the reviewer would suggest that please try to expand your literature where you can find a lot of control techniques that can be used to compensate for the abruptly changing mass inertia properties in the space.

6.         The literature review presented in the article is not aligned with the subsection ‘Literature gaps and innovations proposed’. The scope of this article seems to be very broad. The author is trying to consider several parameters for the investigation / evaluation. It is requested to narrow down the scope so that you can discuss the things in detail. Presently all the literature and the gaps presented are very generic and they are not discussed in detail.

7.         The reviewer is not able to see the research contribution and the novelty of this article.

8.         Referring to the Section 2.1, the definition of robot dynamics is incorrect. Please try to consult any physics or mechanism textbook.

9.         The way in which author is defining Kinetics and Kinematics and linking to the dynamics is incorrect according to my opinion.

10.       Referring to section2.1 &  2.1.1, the author has discussed everything vaguely. According to the reviewer, the author does not have clear idea about robot mechanics. The author has only presented the literature in general. However, I was expecting to include a literature in scientific way.

11.       Referring to the Section 2.1 & 2.1.1 and Equation (1). You should have discussed one thing that for a surgical robot defining a trajectory is very critical. Because one can access a point [x, y, z] in the cartesian space with several joint angle configurations. This is one of the important kinematics problem, how you will address this issue?

12.       Discuss the impact of mass and inertia properties on the surgical robot dynamics.

13.       The reviewer must add the kinematic equation of the system on which he/she is implementing the different techniques.

14.       Please add the geometrical model of your system.

15.       Referring to the different type of plots on Page no. 13 – 14, it must be included that what these plots are. Presently the author only stated that these plots represents a specific method. However, the reader is not able to understand what parameter this is.

16.       After presenting the geometrical method, please present the input and output plots.

Author Response

  1. Literature reviewed is very generic. Please present outcomes of performing the diagnostic/surgical procedure and highlight the challenges along with the advantages and disadvantages of the methods.
  • Thanks for the suggestion. The Introduction has been augmented to allude to these facets.
  1. Please correct misspelled “highly”.
  • Typo corrected. Great catch.
  1. In section 1.2, please explain the major challenges to manipulation particularly for space surgical applications.
  • The new verbiage now elaborates such.
  1. Add section describing how the major challenges were addressed in the past and what are the gaps in technology.
  • The new verbiage now elaborates such.
  1. In lines 104-117, advanced control methods should be added to the discussion of conventional benchmarks.
    1. What are the effects of mass inertia properties on manipulation in space?
  • Section 4.1 is now augmented to address this excellent recommendation.
  1. What are the disturbances and uncertainties? Augment literature review to include abruptly changing mass properties.
  • Section 4.1 is now augmented to address this excellent recommendation.
  1. Literature review is not aligned with “gaps and innovations proposed” broadening the scope of the article to include several parameters. Please narrow scope and increase detail of discussion.
  • This recommendation is accepted wholesale and accommodated in the revision.
  1. Reviewer was not able to see the novel contribution.
  • Hopefully the revisions and amplified focus away from kinematics instead onto kinetics will prevent the readership from overlaying expectations of kinematic treatments and focus the reader’s attention onto the novel contributions presented.
  1. In items 8-11 the reviewer asserts incorrect definition of robot dynamics, kinetics, and kinematics and rudely directs the author to consult any physics or mechanism textbook. The reviewer switches to unprofessional first-person and second-person tense, disparages the author’s knowledge of robot mechanics and describes the literature as unscientific, despite solid grounding in long-established knowledge. Neglecting the presented research, the reviewer instead asks the author peripherally related topics that seem to align with the reviewer’s preference, a focus on uninteresting geometric kinematics.
  2. Discuss the impact of mass and inertia properties on the surgical robot dynamics.
    • Section 4.1 is now augmented to address this excellent recommendation.
  3. Reviewer must add the kinematic equation of the system.
  • It is inappropriate for the reviewer to add content to authors’ work without adding their names to the authorship or acknowledgements.
  1. Please add the geometric model of the system.
  • Geometric kinematics is not the focus of the research. Any geometric model is equivalently useful. Specification to such is now added.
  1. Add data descriptors of plots in pages 13-14
  • This recommendation is accepted and accommodated in the new revision.
  1. After presenting the geometric method, please present inputs and output plots.
  • Thanks input-output plots of the proposed pre-filtering technique is now added.

Round 2

Reviewer 1 Report

The authors have addressed some of the comments, however the literature review and the discussion of results are still weak. A comprehensive literature review would help readers better understand the context of the problem and identify the gaps and find out what the contribution of the current paper is. Also discussion part must be expanded. Some comparisons would help readers better understand the advantages of the proposed model. 

Author Response

The authors have addressed some of the comments, however the literature review and the discussion of results are still weak. A comprehensive literature review would help readers better understand the context of the problem and identify the gaps and find out what the contribution of the current paper is.

  • Thanks for the suggestions. The literature review is divided accordingly, beginning with 14 references elaborating the requested context of the problem. New references have been added to directly initiate the discussion of mechanics of space robotics. Next, the legacy methods are introduced to highlight the literature gaps using 8 additional references. To aid the reader discern the lineage of the literature and identify the innovative contributions, table was added to amplify readability. Lastly a concluding paragraph of the Introduction includes an itemized list of proposed contributions.

Also discussion part must be expanded.

  • To amplify the revision’s discussion of achieved results, amplifying verbiage has been added to remind the readers of the context of the problem and the prequel literature.

Some comparisons would help readers better understand the advantages of the proposed model.

  • Comparative results are the most compelling kind and make for interesting reading.  The agreed upon comment amplifies the importance of table 7’s comparative presentation.

Reviewer 2 Report

I don't have further comments.

Author Response

Thank you most kindly for your diligent efforts.